# The study of GSDMB in pathogenesis of psoriasis vulgaris

Xiaojuan Ji[1]⚬, Huaqing Chen[1]⚬, Ling Xie[1,2,3], Shiqi Chen[1], Shan Huang[1], Qi Tan[1], Huifang Yang[1,2,4], Tao Yang[1,2,4], Xiaoying Ye[1,2,3,4], Zhaolin Zeng[1,2,3,4], Chunlei Wan[1,2,3,4], Longnian Li [1,3,4]*

**1** Department of Dermatology, First Affiliated Hospital of Gannan Medical University, Ganzhou, China, **2** Construction Unit of Branch Center of National Clinical Research Center for Dermatologic and Immunological Diseases, Ganzhou, China, **3** Joint Organization of Jiangxi Clinical Medicine Research Center for Dermatology, Ganzhou, China, **4** Standardized Diagnosis and Treatment Center for, Ganzhou, China

⚬ These authors contributed equally to this work.
* li_longnian@foxmail.com

## Abstract

### Background

Gasdermin (GSDM) B is a member of the GSDM family, which is a protein that may be involved in the cell pyroptosis process and is associated with inflammatory diseases.

### Objective

To explore the correlation between GSDMB and psoriasis vulgaris.

### Methods

Skin lesions from 33 patients with psoriasis vulgaris and 69 normal controls were collected. ELISA and Western blot were adopted to detect proteins. The HaCaT cell line was transfected with 3 sets of interfering sequence siRNA, and the mRNA and protein levels before and after the transfection were measured by qPCR and Western blot respectively, so as to establish a cell model with low *GSDMB* gene expression; the MTT method was used to detect cells viability, flow cytometry to detect cell apoptosis.

### Results

The level of GSDMB protein in the skin lesions of patients with psoriasis vulgaris was lower than that in normal skin tissues ($P < 0.05$). The mRNA and protein expression levels of the target gene in the siRNA-GSDMB-3 group were lower than those in the control group ($P < 0.05$). The proliferation of HaCaT cells was decreased by MTT method and flow cytometry, and the apoptosis rate was increased ($P < 0.05$).

### Conclusion

The expression level of GSDMB in psoriasis vulgaris lesion tissue is lower than that of normal skin tissue. The down-regulation of *GSDMB* expression can inhibit cell proliferation and

**Data Availability Statement:** All relevant data are within the paper and its Supporting Information files.

**Funding:** Funding for this work: Science and Technology Research Project of Education Department of Jiangxi Province. Grant number:

GJJ170887 The funders had no role in study design, data collection and analysis, decision to publish, or preparation of the manuscript.

**Competing interests:** The authors have declared that no competing interests exist.

promote cell apoptosis. GSDMB may play a role in the pathogenesis of psoriasis by affecting the differentiation of keratinocytes and the function of T cells.

# 1. Introduction

Psoriasis is a chronic, inflammatory and recurrent skin disease with a clear boundary of scaly erythema or plaque as the main clinical manifestations [1]. Genome-wide association analysis has confirmed that more than 80 susceptibility loci are associated with psoriasis [2]. Studies have found the psoriasis susceptibility locus rs10852936 [3], and even found a significant correlation between rs10852936 and early-onset psoriasis vulgaris (PV) [4]. Rs10852936 is located in a long-range chromatin interaction region, which includes genes *IKZF3*, *ZPBP2*, *GSDMB*, and *ORMDL3*. Through the review of previous literature, we found that genes *IKZF3*, *ZPBP2*, *GSDMB*, and *ORMDL3* were expressed in lymphocytes, and were significantly associated with the onset of inflammatory and immune diseases such as juvenile asthma [5], Crohn's disease [6] and rheumatoid arthritis [7]. These diseases share some common pathogenesis with psoriasis, which is also an inflammatory and immune disease.

GSDMB is one of six (in humans) gasdermin family members [6]. GSDM family genes have been reported to be involved in regulating the proliferation, differentiation, and apoptosis of epithelial cells, which are related to cancer [8]. As a member of the GSDM family, we believe that GSDMB has similar functions. Previous studies have shown that GSDMB mediates cell pro-death in a non-classical way [9]. However, few reports are on the relationship between keratinocytes and GSDMB [10].

In addition to the abnormal proliferation and apoptosis of keratinocytes, the pathogenesis of psoriasis is also related to the abnormal differentiation of keratinocytes [11]. Therefore, based on the previous data results, we further hypothesized that GSDMB protein may play an important role in the pathogenesis of PV. To test this hypothesis, we examined the expression of GSDMB protein in the skin tissues of patients with PV and normal controls and evaluated its effect on keratinocytes proliferation and apoptosis in vitro. Through this study, we hope to enrich the pathogenesis theory of psoriasis.

# 2. Materials and methods

## 2.1 Materials

Skin of healthy donors and skin lesions of PV patients admitted to the First Affiliated Hospital of Gannan Medical University from January 2019 to October 2020 were collected and frozen in liquid nitrogen for reserve. With patient's consent, skin biopsies collected from patients with a confirmed diagnosis of PV. The control material included skin biopsies collected from healthy volunteers. Collected skin biopsies were stored in liquid nitrogen until the time of the molecular analysis.

The study group included 33 skin biopsies collected from 33 people, 16 women, and 17 men, aged from 10 to 77 years. All patients had a diagnosis of PV confirmed by histopathological examination, and they did not receive psoriasis medication within three months before the histopathological examination. The control group included 69 skin biopsies collected from 69 people, 39 men, and 30 women, aged from 11 to 63 years. No acute infections, diabetes mellitus, hyperlipidemia, atherosclerosis, coronary artery disease, hypertension, cancers, or hormonal disorders was found for each one.

## 2.2 Ethics

The collection and use of sample tissues in this study were informed to the parents/legal guardians of the patients and the patients in advance, and informed consent was signed. Meanwhile, the collection of samples was approved by the Ethics Committee of the Gannan Medical University.

## 2.3 Methods

**2.3.1 Cell culture.** HaCaT cell line, BNCC101683, purchased from BNA Bio. Culture conditions: The medium was Dulbecco's Modified Eagle Medium (DMEM) and 10% North American fetal bovine serum, PH was kept at 7.3, and the medium was placed in a 37°C constant temperature incubator with a $CO_2$ concentration of 5% and humidity of 70%-80%. The cells grew adherent monolayer, and when they grew to 80%-90% confluence, they were diluted into different cell densities for cell passage and subsequent experiments according to the experimental purpose.

**2.3.2 ELISA.** The expression of target protein was detected by ELISA double antibody sandwich method in 21 psoriasis patients and 57 normal controls. The tissue protein was extracted and the optical density (OD) of each well at 450 nm was measured by ELISA. Draw the standard curve: Take the concentration of standard substance as abscissa and the measured absorbance OD value as ordinate, draw the standard curve with Excel software, fit the corresponding regression equation and correlation coefficient $R^2$, and calculate the true level of GSDMB in the sample.

**2.3.3 Western blot.** Western blot was used to detect the expression of target protein in the skin lesions of 12 patients with PV and 12 normal controls, and the experimental group and the control group were randomly paired and grouped. The total protein of the sample was obtained by grinding the tissue, and the cell was lysed to extract the cell protein for quantification. Samples were taken from 20 μg per well using glyceraldehyde-3-phosphate dehydrogenase (GAPDH) as an internal reference and transferred to the normal control (NC) membrane by sodium dodecyl sulfate-polyacrylamide gel electrophoresis (SDS-PAGE). The proteins were then stored in 5% skim milk powder at room temperature for 1 h, incubated overnight in Rabbit anti-GSDMB, and washed with tris-buffered saline + 0.1% [v/v] Tween-20 (TBST) (5×) three times for about 10 min every time. The antibody 2 (horseradish enzyme-labeled Goat anti-rabbit IgG (H+L)) was incubated at room temperature for 1 h, washed with TBST three times for about 10 min every time, and the developer solution was applied for external application. The antibody bands were exposed and photographed by a gel imager, and the gray values of each antibody band were analyzed by software.

**2.3.4 Transient HaCaT cell transfection.** Add liposome-medium mixture to siRNA-medium mixture, mix well and leave at room temperature for 15 min. Transfection can be carried out when cell density reaches 70–80%. The transfection reagent was added to the 6-well plate, and after cross shaking, the transfection reagent was placed into the incubator for cultivation (cultivation conditions were the same as before) for 4 h. After 4 h, the medium was replaced and the culture was continued for 24–48 h. Twenty-four hours later, the culture plates were taken out and the cells were detected by real-time fluorescence quantitative PCR. Western blot, MTT, and flow cytometry were performed 48 h later.

**2.3.5 Quantitative real-time PCR, qPCR.** Total RNA was extracted from the sample using Trizon Reagent (CW0580S, CWBIO); then, first-strand complementary DNA was synthesized with the use of HiFi Script Reagent Kit (CW2569M, CWBIO RT). After that, qPCR was carried out with 2×SYBR Green PCR Master Mix (Xiamen Life Internet Technology Co. LTD). The whole RT-qPCR assay procedures were conducted in accordance with the manufacturer's instructions. *GAPDH* was selected as reference gene.

**2.3.6 Cell proliferation detection (3-(4,5)-dimethylthiahiazo (-z-y1)-3,5-di-phenytetra-zoliumromide, MTT).**   A volume of 200 μL cell suspension was transferred to 96-well culture plates with $5 \times 10^3$ cells in each well for 3–5 days. Add 20 μL MTT reagent to each well, mix well to avoid light, and continue to incubate for 4 h (MTT 5 mg/mL, PH 7.4). Siphon off the old culture solution and add 100 μL dimethyl sulfoxide (DMSO) to the mixture, place on the shaker for 10 min, and mix well. The wavelength of ELISA was set at 490 nm, the absorbance value of each group of cells was read and the results were recorded.

**2.3.7 Apoptosis detection (Flow Cytometry).**   The HaCaT cell culture plate was taken out and 500 μL trypsin without ethylene diamine tetraacetic acid (EDTA) was added for centrifugation at about 2 min to remove the supernatant. Phosphate buffered saline (PBS) was added for rinsing and centrifugation to absorb the bleaching solution. A volume of 1 mL of precooled 70% ethanol added to each tube and fixed at 4˚C for more than 2 h. Centrifugation was performed at 8000 r/min for 3 min, and the fixative was discarded. Add 1 mL 1× PBS to each tube, blow and mix, centrifuge at 8000 r/min for 3 min, and discard PBS. A volume of 100 μL of 1× binding buffer was added to resuscitate cells. Staining with Annexin V-FITC/PI Apoptosis Kit (Hangzhou Lianke Biotechnology Co. LTD). Mix well and keep away from light for 10 min at room temperature. The up-flow cytometry was performed within 1 h and the data were analyzed.

**2.3.8 Data analysis and statistical methods.**   IBM SPSS 20.0 and GraphPad Prism 6.0 were used for statistical analysis and image making for all data in this study. Measurement data were expressed as $\bar{X} \pm S$, t-test was used for comparison between two independent samples, one-way ANOVA was used for comparison between multiple groups, and SNK-Q test was used for comparison within groups. The test levels $\alpha = 0.05$, and $P < 0.05$ were considered statistically significant.

## 3. Results

### 3.1 The general information

The two groups were comparable in gender and age ($P > 0.05$), as shown in Table 1.

### 3.2 The expression of GSDMB protein in the skin of patients with PV was decreased

We used ELISA to detect the expression of GSDMB protein in the skin of 78 participants (21 PV patients, 57 normal subjects) and Western blot to detect the expression of GSDMB protein in the skin of 24 participants (12 PV patients, 12 normal subjects) (Tables 2 and 3). The results of both tests showed that the expression level of GSDMB in PV patients was lower than that in normal skin tissues, and the difference was statistically significant (t = 2.83, $P < 0.05$) (Fig 1A and 1B). Therefore, the low expression of GSDMB may be related to the development of PV.

**Table 1.  Data of experimental group and control group were compared.**

| Group | Number | Age range (Year) | Average age (Year) | Male/female |
|---|---|---|---|---|
| Experimental | 33 | 10–77 | 41.09±16.17 | 17/16 |
| Control | 69 | 11–63 | 37.90±13.59 | 39/30 |
| $t/\chi^2$ | / | / | 1.04 | 0.22 |
| P | / | / | 0.30 | 0.63 |

**Table 2. ELISA method to detect the protein level of GSDMB in skin tissue (ng/mL).**

| Group | Number | Mean value | Standard deviation | t | P |
|---|---|---|---|---|---|
| Normal | 57 | 17.88 | 1.53 | 2.83 | 0* |
| PV | 21 | 16.76 | 1.54 | | |

PV group compared with normal group

*$P < 0.05$.

**Table 3. Western blot detection of the relative protein level of GSDMB in skin tissue.**

| Group | Number | Mean value | Standard deviation | t | P |
|---|---|---|---|---|---|
| Normal | 12 | 1.84 | 0.94 | 3.59 | 0* |
| PV | 12 | 0.67 | 0.61 | | |

Pv group compared with normal group

*$P < 0.05$.

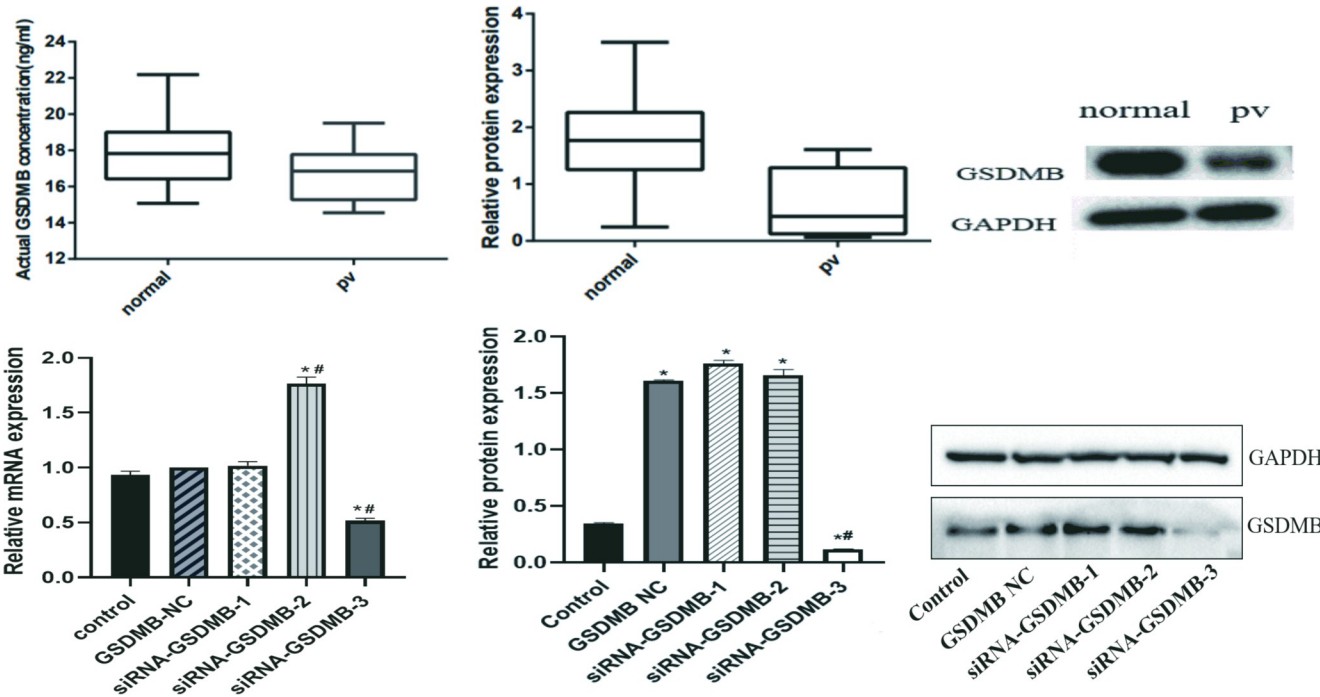

**Fig 1. GSDMB in pathogenesis of psoriasis vulgaris.** (A) GSDMB levels in skin tissues detected by ELISA. PV group compared with normal group, *$P < 0.05$. (B) GSDMB levels in skin tissues detected by Western blot. PV group compared with normal group, *$P < 0.05$. (C) mRNA expression levels in each group were detected by qPCR. Compared with the control group, *$P < 0.05$; compared with GSDMB-NC, #$P < 0.05$. (D) Western blot was used to detect the protein expression levels in each group. Compared with the control group, *$P < 0.05$; compared with GSDMB-NC, #$P < 0.05$.

### 3.3 The mRNA expression of *GSDMB* decreased in group siRNA-GSDMB-3 cells

SiRNA of 3 items of gene *GSDMB* and one non-specific negative control siRNA were transfected into HaCaT cells respectively. Using *GAPDH* as an internal reference, qPCR detection of the relative expression level of mRNA in the siRNA-GSDMB-1 group, siRNA-GSDMB-2

**Table 4. qPCR detection of the relative expression level of mRNA in each group of cells.**

| Comparison between groups | The difference between the two means | Standard error | P |
|---|---|---|---|
| si-1 with control | -0.83 | 0.02 | 0.35 |
| si-2 with control | -0.83 | 0.03 | 0 * |
| si-3 with control | 0.41 | 0.02 | 0* |
| NC with control | -0.06 | 0.01 | 0.53 |
| si-1 with NC | -0.01 | 0.02 | 0.99 |
| si-2 with control | -0.76 | 0.03 | 0.01# |
| si-3 with control | 0.48 | 0.01 | 0# |

Compared with the control group

* $P < 0.05$, compared with GSDMB-NC

# $P < 0.05$.

group, siRNA-GsdMB-3 group, *GSDMB* gene interference no-load group (GSDMB-NC), and normal control group (control) of cells. Compared with the control group, the mRNA expression of *GSDMB* in the si-3 group was significantly decreased, and the difference was statistically significant (Table 4 and Fig 1C). The results indicated that the siRNA-GSDMB-3 group could successfully construct *GSDMB* low expression model.

### 3.4 The protein expression of GSDMB decreased in group siRNA-GSDMB-3 cells

Five siRNA interference sequences were transfected into HaCaT cells, and total proteins were extracted 48 h later. Western blot detection of the relative expression level of GSDMB protein in each group of cells. The ratio of GSDMB gray value to GAPDH gray value was used to represent the relative expression level of GSDMB protein. The results were shown in Table 5, and Fig 1D. The expression of GSDMB protein in the si-3 group was significantly lower than that in the control group and no-load group ($P < 0.05$). The above experiments indicate that the interfering sequence siRNA-GSDMB-3 can be successfully transfected into HaCaT cells and achieve low expression.

### 3.5 Down-regulated expression of GSDMB promoted apoptosis of HaCaT cells

The apoptosis of the transfected HaCaT cells in the si-3 group, NC group, and control group was detected by flow cytometry respectively. The results showed that compared with the

**Table 5. Western blot detection of the relative expression level of GSDMB protein in each group of cells.**

| Comparison between groups | The difference between the two means | Standard error | P |
|---|---|---|---|
| si-1 with control | -1.41 | 0.01 | 0* |
| si-2 with control | -1.31 | 0.02 | 0* |
| si-3 with control | 0.22 | 0.00 | 0* |
| NC with control | -1.26 | 0.01 | 0* |
| si-1 with NC | -0.15 | 0.01 | 0.06 |
| si-2 with NC | -0.05 | 0.02 | 0.91 |
| si-3 with NC | 1.49 | 0.00 | 0# |

Compared with the control group

* $P < 0.05$; compared with GSDMB-NC

# $P < 0.05$.

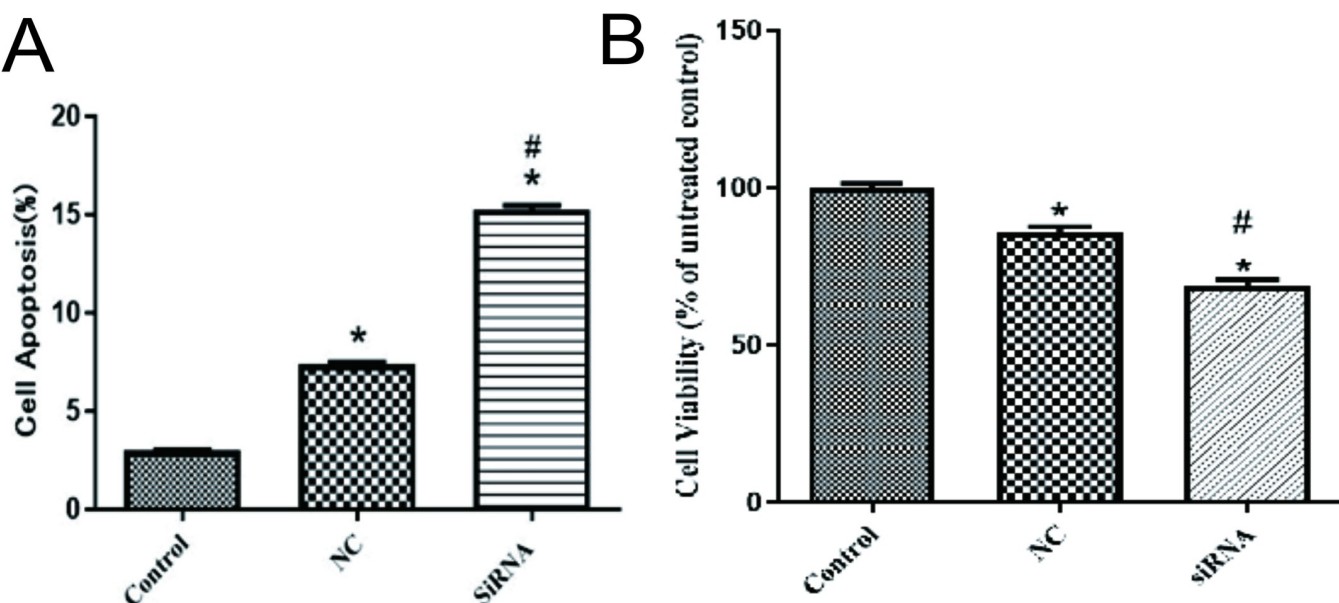

**Fig 2. Proliferation and apoptosis of HaCaT cells.** (A) Cell apoptosis was detected by flow cytometry. Compared with the control group, $^{*}P < 0.05$, compared with NC, $^{#}P < 0.05$. (B) Cell proliferation was detected by MTT assay. Compared with the control group, $^{*}P < 0.05$; compared with NC, $^{#}P < 0.05$.

control group and *GSDMB* gene interference no-load group, the interference group (siRNA) had the highest apoptosis rate (Fig 2A and Table 6).

## 3.6 The expression of *GSDMB* was down-regulated and the proliferation of HaCaT cells was inhibited

The transfected HaCaT cells were inoculated into the culture plate with a density of 5000 cells per well and a volume of 200 μL per well. The proliferation of the si-3 group, NC group, and control group were detected by the MTT method. MTT results was shown in Fig 2B, compared with the control group and the NC group, the interference group had the lowest proliferation ability ($P < 0.05$).

## 4. Discussion

Previous studies have shown that the SNP rs10852936 in 17q12 is associated with PV, and the *GSDMB* gene is located in 17q12-q21 [4, 12]. The pathogenesis of PV and GSDMB has not been reported, but genes in this region have been associated with immune-mediated inflammatory diseases (IMID) such as asthma [13, 14]. Evidence continues to show that IMID and

**Table 6. Apoptosis rate detected by flow cytometry (%).**

| Comparison between groups | The difference between the two means | Standard error | P |
|---|---|---|---|
| si-3 with control | -12.25 | 0.2 | 0* |
| NC with control | -4.37 | 0.1 | 0* |
| si-3 with NC | -7.88 | 0.22 | 0# |

Compared with the control group

$^{*}P < 0.05$; compared with GSDMB-NC

$^{#}P < 0.05$.

psoriasis share many common inflammatory pathways [15]. In this study, we found that the expression of GSDMB in the skin tissue of PV patients was lower than that of normal people, indicating that GSDMB may play a role in the occurrence of PV.

The pathogenesis of psoriasis is related to abnormal differentiation of keratinocytes such as hyperkeratinization and insufficiency [16]. In normal skin, keratinocytes continuously differentiate and mature from basal layer to stratum corneum, and in the process of differentiation and maturation, keratinocytes transform from nucleated cells to non-nucleated cells, from organelles integrity to organelles disappearance, and finally form scales through the stratum corneum. At this point, the terminal differentiation of keratinocytes reaches its peak in the stratum corneum. In psoriatic lesions, the cell cycle of keratinocytes is significantly shortened, and the differentiation of granular layer and stratum corneum is also accelerated with the acceleration of cell growth. Therefore, the phenomenon of residual nucleus can still be seen in the terminal differentiation stage of keratinocytes, namely, insufficiency of keratinocytes, which is the manifestation of abnormal differentiation.

Studies have shown that GSDMB is involved in the regulation of cell proliferation and apoptosis [6, 17], down-regulating the expression of GSDMB weakens cell proliferation [18]. In order to explore the role of down-regulation of GSDMB in this process, we successfully down-regulated the expression of GSDMB in HaCaT cells. Flow cytometry and MTT assay were used to detect cell apoptosis and proliferation, respectively. It was found that down-regulation of GSDMB promoted the apoptosis of HaCaT cells and inhibited the proliferation of HaCaT cells.

Overdevelopment of keratinocytes toward differentiation inhibits their own proliferation [19], so factors that promote keratinocyte proliferation may be viewed as preventing differentiation. Therefore, downregulation of GSDMB gene, which is involved in cell proliferation, is likely to play a role in the accelerated differentiation of keratinocytes in the granular layer and stratum corneum of psoriasis. The keratinization process can only begin after the cells have undergone certain differentiation [20]. Studies have shown that keratinization is related to the expression of GSDMB [10, 21], and the expression level of GSDMB is up-regulated during the terminal differentiation of human keratinocytes in vitro [10], while the expression of GSDMB is deficient in animals with epidermal keratinization deficiency [10]. Therefore, skin insufficiency of psoriasis patients may be closely related to the downregulation of GSDMB expression.

Current studies on the pathogenesis of psoriasis mainly focus on the theory of T-cell-mediated immunity [22, 23]. Evidence has shown that the IL-23/IL-17 axis plays a pivotal role in inflammatory events leading to psoriasis, IMID and other manifestations [24, 25]. Psoriasis not only shares the same inflammatory pathway with IMID, but also has been found to share the same 17q21 risk site with various other IMID, and it has been shown that T cells are one of the cell types most significantly affected by the 17q21 variant [26]. All the above evidences indicate that *GSDMB* located at 17q12-q21 may be related to psoriasis, and the pathogenesis may involve abnormal regulation of T cells. Future studies can focus on the correlation between GSDMB and abnormal T cell function in psoriasis lesions.

Our results suggest that the downregulation of *GSDMB* may be involved in the abnormal keratinization of psoriasis, and provide insights for further study of the association between psoriasis and GSDMB. Our results suggest that the down-regulated expression of *GSDMB* gene in psoriatic lesions is a suitable model for future studies of the relationship between psoriasis and GSDMB. However, *GSDMB* is to some extent unique to the human genome [6], so the effects of GSDMB on psoriasis proliferation and differentiation have not been tested in animal models. The limited number of specimens and the lack of establishment of in vivo models are the shortcomings of this study. The study of GSDMB in the pathogenesis of psoriasis vulgaris remains to be further explored.

## 5. Conclusion

By comparing the differential expression of GSDMB in the skin tissues of normal people and psoriasis patients, this study suggested that GSDMB might be related to psoriasis. Further-more, siRNA interference sequences were constructed, and the effects of down-regulation of *GSDMB* on HaCaT cell proliferation and apoptosis were detected by MTT and flow cytometry. The results showed that down-regulation of *GSDMB* expression could promote apoptosis and inhibit cell proliferation. The gene *GSDMB* may play a role in the pathogenesis of psoriasis by influencing keratinocyte differentiation and the function of T cells in skin lesions.

## Supporting information

**S1 File.**
(PDF)

## Author Contributions

**Conceptualization:** Xiaojuan Ji, Huaqing Chen, Longnian Li.

**Data curation:** Longnian Li.

**Funding acquisition:** Chunlei Wan.

**Investigation:** Xiaojuan Ji, Huaqing Chen, Ling Xie, Shiqi Chen, Shan Huang, Qi Tan, Hui-fang Yang, Tao Yang, Xiaoying Ye, Zhaolin Zeng, Chunlei Wan.

**Project administration:** Longnian Li.

**Resources:** Zhaolin Zeng.

**Software:** Chunlei Wan.

**Supervision:** Chunlei Wan, Longnian Li.

**Validation:** Longnian Li.

**Writing – original draft:** Xiaojuan Ji, Huaqing Chen, Ling Xie, Shiqi Chen.

**Writing – review & editing:** Longnian Li.

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
