## [Decision Letter · Decision Letter 0]

31 Oct 2022

PONE-D-22-26732The study of GSDMB in pathogenesis of psoriasis vulgarisPLOS ONE

Dear Dr. Li,

Thank you for submitting your manuscript to PLOS ONE. After careful consideration, we feel that it has merit but does not fully meet PLOS ONE’s publication criteria as it currently stands. Therefore, we invite you to submit a revised version of the manuscript that addresses the points raised during the review process. Please carefully address reviewer 3's comments and consider proper additional experiments. 

We look forward to receiving your revised manuscript.

Kind regards,

Yi Cao

Academic Editor

PLOS ONE

Journal Requirements:

5. Please upload a new copy of Figure 1 as the detail is not clear. Please follow the link for moreinformation:

https://blogs.plos.org/plos/2019/06/looking-good-tips-for-creating-your-plos-figures-graphics/

https://blogs.plos.org/plos/2019/06/looking-good-tips-for-creating-your-plos-figures-graphics/

6. We note you have included a table to which you do not refer in the text of your manuscript. Please ensure that you refer to Table 2, 3 and 6 in your text; if accepted, production will need this reference to link the reader to the Table.

Reviewers' comments:

Reviewer's Responses to Questions

**Comments to the Author**

1. Is the manuscript technically sound, and do the data support the conclusions?

Reviewer #1: Yes

Reviewer #2: Partly

Reviewer #3: No

2. Has the statistical analysis been performed appropriately and rigorously? 

Reviewer #1: Yes

Reviewer #2: Yes

Reviewer #3: Yes

3. Have the authors made all data underlying the findings in their manuscript fully available?

Reviewer #1: Yes

Reviewer #2: No

Reviewer #3: No

4. Is the manuscript presented in an intelligible fashion and written in standard English?

Reviewer #1: Yes

Reviewer #2: Yes

Reviewer #3: Yes

5. Review Comments to the Author

Reviewer #1: Xiaojuan Ji et al. performed an interesting study on the role of GSDMB in the pathogenesis of psoriasis. However, there are several issues that should be addressed.

Abstract

Line 25 “GSDMB is a member of the GSDM (gasdermin) family” should be reformulated as follows “Gasdermin (GSDM) B is a member of the GSDM family”

Introduction

Line 50 The abbreviation PS for psoriasis is not useful. The abbreviation PV for psoriasis vulgaris is sufficient.

Line 68 “However, few report is about the relationship between keratinocytes and GSDMB”. The authors should provide bibliographic references and improve grammar, I suggest 'few reports are on'.

Line 70 “abnormal differentiation of hyperkeratosis” Psoriasis is characterized by abnormal differentiation of keratinocytes. Line 71 “incomplete keratosis”. Psoriasis is characterized by parakeratosis that refers to incomplete maturation of epidermal keratinocytes. I consider that this phrase is misleading and the authors should rephrase it.

Materials and methods

Line 80 “Skin lesions or skin of PV patients …” This statement is misleading.

Results

All tables should be mentioned in the text. The resolution of Figure 1 should be improved.

Discussion

When authors use expressions such as 'studies have shown', they should indicate several references, not just one (e.g. line 237, line 257, etc). Please review the entire manuscript.

Line 238 “GSDMB located at 17q12-21”. The authors mention it again at line 249.

Line 240 “There is growing evidence that there is a correlation between immune-mediated inflammatory diseases and psoriasis”. The authors should provide bibliographic references.

Line 274, 278 – The reference number should be moved to the end of the sentence.

The English language should be revised.(e.g. line 253-255).

Reviewer #2: The authors investigated the role of GSDMB in the pathogenesis of psoriasis. This brings new insights into the pathophysiology of the psoriasis and clinical investigation of the GSDMB. Here are some comments to improve the impact of the article.

Minor comment:

1) The author mentioned GSDMB’s role in cell pyrolysis process (Line 26), it will better to clary the potential effects of pyrolysis on psoriasis, since there are no reports on pyrolysis in psoriasis. Do you mean proptosis? If so, please correct it.

2) Line 102, CO2 should be CO2

3) Lines 265-266, the author said “GSDMB is not involved in the pathogenesis of psoriasis by affecting keratinocyte proliferation/apoptosis”, but the results showed GSDMB can promote HaCaT proliferation and inhibit apoptosis. Can author verify the results and conclusion?

4) References 12, 22, and 24, the author names are capital, inconsistent with other ones.

5) The figures are low-quality, please replace higher ones.

Reviewer #3: In this manuscript, the authors demonstrate that The level of GSDMB protein in the skin lesions of patients with psoriasis vulgaris was lower than that in normal skin tissues and demonstrate that the down-regulation of GSDMB expression can inhibit cell proliferation and promote cell apoptosis.

However, psoriasis is a kind of inflammatory disease, if you don’t test function of this gene in inflammation, we can’t tell its importance in this disease. What’s more, i do think you should build the psoriasis model in your experiments to test the gene function.

6. PLOS authors have the option to publish the peer review history of their article (what does this mean?). If published, this will include your full peer review and any attached files.

Reviewer #1: No

Reviewer #2: No

Reviewer #3: No

---

## [Author Response · Author response to Decision Letter 0]

12 Dec 2022

Reviewer 1:

Comment 1: Line 25 “GSDMB is a member of the GSDM (gasdermin) family” should be reformulated as follows “Gasdermin (GSDM) B is a member of the GSDM family”

Response: We are so grateful for your kind advice. Considering the reviewer’s suggestion, we have rewritten this part as “Gasdermin (GSDM) B is a member of the GSDM family”.

Comment 2: Line 50 The abbreviation PS for psoriasis is not useful. The abbreviation PV for psoriasis vulgaris is sufficient.

Response: Thanks so much for your kind suggestion. In reference to the reviewer's suggestion, the abbreviation PS for Psoriasis has been removed from the sentence.

Comment 3: Line 68 “However, few report is about the relationship between keratinocytes and GSDMB”. The authors should provide bibliographic references and improve grammar, I suggest 'few reports are on'.

Response: We are so grateful for your kind advice. Considering the reviewer’s suggestion, we have rewritten this part as “few reports are on the relationship between keratinocytes and GSDMB” and add an reference [11 ] at the end of this sentence.

Comment 4: Line 70 “abnormal differentiation of hyperkeratosis” Psoriasis is characterized by abnormal differentiation of keratinocytes. Line 71 “incomplete keratosis”. Psoriasis is characterized by parakeratosis that refers to incomplete maturation of epidermal keratinocytes. I consider that this phrase is misleading and the authors should rephrase it.

Response: Thanks so much for your kind suggestion. In reference to the reviewer's suggestion, we delete “incomplete keratosis” and rewrite this part as “abnormal differentiation of keratinocytes”.

Comment 5: Line 80 “Skin lesions or skin of PV patients …” This statement is misleading.

Response: Thanks for your suggestion. In reference to your kind suggestion,, the statements of “Skin lesions or skin of PV patients” are corrected as “Skin of healthy donors and skin lesions of PV patients”.

Comment 6: All tables should be mentioned in the text.

Response: It is really true as your suggestion that all tables should be mentioned in the text. So, we make some changes: Line 177-178, mentions about table 2 and 3, line 220, mentions about table 6 are added.

Comment 7: The resolution of Figure 1 should be improved.

Response: Considering the reviewer’s suggestion, In order to improve the clarity of the picture, we have divided the picture into two parts: Fig 1 and Fig 2.

Comment 8: Line 238 “GSDMB located at 17q12-21”. The authors mention it again at line 249.

Response: Thanks for your suggestion, we have deleted the redundant description of “located at 17q12-21” at line 249.

Comment 9: The English language should be revised.(e.g. line 253-255).

Response: Thanks for the reviewer's suggestion, the sentence of “GSDMB expression in PV patients' skin tissues was lower than that of normal controls, which indicate that GSDMB may play a role in PV occurrence, and this result is consistent with our expected hypothesis” is reformulated as “the expression of GSDMB in the skin of PV patients was lower than that of normal controls, which indicate that GSDMB may play a role in the pathogenesis of PV, and this result is consistent with our hypothesis.”(line 253-255)

Comment 10: Line 274, 278 – The reference number should be moved to the end of the sentence.

Response: Special thanks for your comments.We have moved the reference number to the end of the sentence.( Line 274, 278)

Reviewer 2: 

Comment 1: The author mentioned GSDMB’s role in cell pyrolysis process (Line 26), it will better to clary the potential effects of pyrolysis on psoriasis, since there are no reports on pyrolysis in psoriasis. Do you mean proptosis? If so, please correct it.

Response: Thank you for pointing out the problem with this word. Considering the meaning of the passage, we have corrected “pyrolysis” to "pyroptosis”.

Comment 2: Line 102, CO2 should be CO2.

Response: Special thanks for pointing out our mistakes., The statement of “CO2” is corrected to “CO2”.

Comment 3: Lines 265-266, the author said “GSDMB is not involved in the pathogenesis of psoriasis by affecting keratinocyte proliferation/apoptosis”, but the results showed GSDMB can promote HaCaT proliferation and inhibit apoptosis. Can author verify the results and conclusion?

Response: Thank you very much for the questions you pointed out. We are sorry that this part was not clear in the original manuscript. By referring to relevant literature and materials, we have revised the “discussion part” of our manuscript. We hope the following sentences can get your support: “Our results and related studies have shown that GSDMB is involved in the regulation of cell proliferation and apoptosis [19, 20], down-regulating the expression of GSDMB weakens cell proliferation [21]. Overdevelopment of keratinocytes toward differentiation inhibits their own proliferation [22], so factors that promote keratinocyte proliferation can be viewed as preventing differentiation. Therefore, downregulation of GSDMB gene, which is involved in cell proliferation, is likely to play a role in the accelerated differentiation of keratinocytes in the granular layer and stratum corneum of psoriasis.”

[19]. Zheng Z, Deng W, Lou X, Bai Y, Wang J, Zeng H, et al. Gasdermins: pore-forming activities and beyond. Acta Biochim Biophys Sin (Shanghai). 2020; 52(5):467-74. https://doi.10.1093/abbs/gmaa016 PMID: 32294153

[20]. Zou J, Zheng Y, Huang Y, Tang D, Kang R, Chen R. The Versatile Gasdermin Family: Their Function and Roles in Diseases. Front Immunol. 2021; 12:751533. https://doi. 10.3389/fimmu.2021.751533 PMID: 34858408

[21]. Fu L, Bao J, Li J, Li Q, Lin H, Zhou Y, et al. Crosstalk of necroptosis and pyroptosis defines tumor microenvironment characterization and predicts prognosis in clear cell renal carcinoma. Front Immunol. 2022; 13:1021935. https://doi. 10.3389/fimmu.2022.1021935 PMID: 36248876

[22]. Molinuevo R, Freije A, Contreras L, Sanz JR, Gandarillas A. The DNA damage response links human squamous proliferation with differentiation. J Cell Biol. 2020; 219(11):e202001063. https://doi. 10.1083/jcb.202001063 PMID: 33007086

Comment 4: References 12, 22, and 24, the author names are capital, inconsistent with other ones.

Response: Special thanks for your kind suggestion on the refereces, we have corrected the uppercase author names to lowercase.

Reviewer 3: 

Comment: In this manuscript, the authors demonstrate that The level of GSDMB protein in the skin lesions of patients with psoriasis vulgaris was lower than that in normal skin tissues and demonstrate that the down-regulation of GSDMB expression can inhibit cell proliferation and promote cell apoptosis.

However, psoriasis is a kind of inflammatory disease, if you don’t test function of this gene in inflammation, we can’t tell its importance in this disease. What’s more, i do think you should build the psoriasis model in your experiments to test the gene function.

Response: Thank you for your valuable advice. Our results suggest that the down-regulated expression of GSDMB gene in psoriatic lesions is a suitable model for future studies of the relationship between psoriasis and GSDMB. However, GSDMB is to some extent unique to the human genome [30], so the effects of GSDMB on psoriasis proliferation and differentiation have not been tested in animal models. The lack of establishment of in vivo models is the shortcomings of this study. We will try our best to further explore the role of GSDMB in the pathogenesis of psoriasis. Thank you again for your suggestions.

[30]. Zou J, Zheng Y, Huang Y, Tang D, Kang R, Chen R. The Versatile Gasdermin Family: Their Function and Roles in Diseases. Front Immunol. 2021; 12:751533. https://doi. 10.3389/fimmu.2021.751533 PMID: 34858408

The article has been revised as required and any additional material has been submitted as supporting information. We look forward to hearing from you regarding our submission. We would be glad to respond to any further questions and comments that you may have. Appreciate for Editor/Reviewers.

---

## [Decision Letter · Decision Letter 1]

19 Dec 2022

The study of GSDMB in pathogenesis of psoriasis vulgaris

PONE-D-22-26732R1

Dear Dr. Li,

We’re pleased to inform you that your manuscript has been judged scientifically suitable for publication and will be formally accepted for publication once it meets all outstanding technical requirements.

Kind regards,

Yi Cao

Academic Editor

PLOS ONE

Additional Editor Comments (optional):

Reviewers' comments:

Reviewer's Responses to Questions

**Comments to the Author**

1. If the authors have adequately addressed your comments raised in a previous round of review and you feel that this manuscript is now acceptable for publication, you may indicate that here to bypass the “Comments to the Author” section, enter your conflict of interest statement in the “Confidential to Editor” section, and submit your "Accept" recommendation.

Reviewer #1: All comments have been addressed

Reviewer #2: All comments have been addressed

2. Is the manuscript technically sound, and do the data support the conclusions?

Reviewer #1: Yes

Reviewer #2: Yes

3. Has the statistical analysis been performed appropriately and rigorously? 

Reviewer #1: Yes

Reviewer #2: Yes

4. Have the authors made all data underlying the findings in their manuscript fully available?

Reviewer #1: Yes

Reviewer #2: Yes

5. Is the manuscript presented in an intelligible fashion and written in standard English?

Reviewer #1: Yes

Reviewer #2: Yes

6. Review Comments to the Author

Reviewer #1: Manuscript has been adequately improved. The authors have addressed all suggestions; I consider the manuscript could be published, according to the Editor's choice.

Reviewer #2: The authors addressed all the comments. There are no additional comments for the author or any concern about dual publication, research ethics, or publication ethics.

7. PLOS authors have the option to publish the peer review history of their article (what does this mean?). If published, this will include your full peer review and any attached files.

Reviewer #1: No

Reviewer #2: No

---

## [Editor Report · Acceptance letter]

27 Dec 2022

PONE-D-22-26732R1 

The study of GSDMB in pathogenesis of psoriasis vulgaris 

Dear Dr. Li:

I'm pleased to inform you that your manuscript has been deemed suitable for publication in PLOS ONE. Congratulations! Your manuscript is now with our production department. 

Kind regards, 

on behalf of

Dr. Yi Cao 

Academic Editor

PLOS ONE